# Beam-Reconfigurable Multi-Antenna System with Beam-Combining Technology for UAV-to-Everything Communications

**Yu-Seong Choi** , **Jeong-Su Park** and **Wang-Sang Lee** *

Department of Electronic Engineering/Engineering Research Institute (ERI), Gyeongsang National University (GNU), 501, Jinju-daero, Jinju-si, Gyeongsangnam-do 52828, Korea; cys4140@gnu.ac.kr (Y.-S.C.); bjs100095@gnu.ac.kr (J.-S.P.)
* Correspondence: wsang@gnu.ac.kr; Tel.: +82-55-772-1728

**Abstract:** This paper proposes a beam-reconfigurable antenna for unmanned aerial vehicles (UAVs) with wide beam coverage by applying beam-combining technology to multiple antennas with different beam patterns. The proposed multi-antenna system consists of a circular patch antenna and a low-profile printed meandered monopole antenna. For beam combining, a coplanar waveguide with ground (CPW-G) structure feeding network is proposed, and it consists of two input ports, a 90° hybrid coupler, a microstrip 90° phase delay line, and a single-pole double-throw (SPDT) switch. It performs the role of power distribution and phase adjustment, and synthesizes the broad-side beam of the monopole antenna and the end-fire beam of the patch antenna to form the directive broadside beams in four different directions. The proposed antenna system operates at 5–5.5 GHz which covers both UAV ground control frequencies (5.03–5.09 GHz) and UAV mission frequencies (5.091–5.150 GHz). The peak gain, total efficiency, and half-power beamwidth (HPBW) of the antenna system are approximately 5.8 dBi, 76%, 145° in the elevation plane, and 360° in the azimuth plane respectively. Its electrical size and weight are $\lambda_0 \times \lambda_0 \times 0.21\lambda_0$ at 5.09 GHz and 19.2 g, respectively.

**Keywords:** antennas; antenna arrays; beam-combining; beam-reconfigurable; multi-antenna system; UAV antenna; UAV to everything; wide coverage

## 1. Introduction

In the past, unmanned aerial vehicles (UAVs) were developed for military use, but recently they can be used in various applications such as weather observation, remote farm management, lifesaving and filming, disaster monitoring, telecommunications link, and leisure. It is rapidly expanding into the private and commercial markets. UAV wireless communication technology is essential when performing missions with remote or automatic control, and robust UAV-to-everything (U2X) communication technology is required for swarming flight and beyond visual line of sight (BVLoS) operations. Therefore, the proposed beam-reconfigurable multi-antenna system with a lightweight, aerodynamic minimization structure, and wide operating coverage is essential.

Due to the increased UAV market size and usability, the analysis of UAV-to-infra [1] and UAV-to-UAV [2] communication environments for U2X communication has been conducted, and various UAV antennas have been proposed [3–10]. In [3], the proposed microstrip magnetic dipole antenna is attached to the landing gear of a multicopter drone. Although the antenna has space efficiencies in the antenna placement, it has a large size and a null due to the characteristics of the conical beam. Refs. [4,5] have a low-profile small size, but due to a limited beam coverage, they are not suitable for performing robust U2X applications. Refs. [6,7] operate in the frequency band (5.03–5.15 GHz)

dedicated to UAVs and can achieve wide elevation angle using the configuration of the proposed feeding network. However, it is difficult to provide hemispheric beam coverage. In addition, the rectangular box-shaped structure is not suitable for a structure for minimizing aerodynamic resistance. Refs. [8–10] have high gain characteristics exploiting pattern multiplication, but the beam coverage is limited. In particular, Ref. [10] has a wide beam coverage, but a fade zone exists near the elevation angle of 0°, where the beampattern synthesis is perpendicular from the center of the antenna array. In [11], the antenna has a low-profile characteristic, but the antenna configuration is complex and its weight is heavy due to metal substrates. It is characterized by the ability to tilt the beam in the desired direction using the adjustment of diodes or a feeding network, but several modes can make antenna beam control difficult [12–14]. In this paper, we proposed a beam-reconfigurable multi-antenna system with beam-combining technology for U2X communications of UAVs. The proposed multi-antenna system with different beams is electronically controlled by the proposed feeding network, releasing the reconfigurable beams in four different directions. When the areas covered by these beams are combined, they achieve a hemispherical coverage. In addition, the proposed antenna is suitable for UAV flight due to its simple structure and a light-weight characteristic, and it covers both UAV ground control frequency (5.03–5.09 GHz) and UAV mission frequency (5.091–5.15 GHz) bands. Therefore, it is essential to establish a robust U2X communication technology by applying it to various UAV applications.

## 2. Operating Principle and Antenna System Configuration

Figure 1 is a schematic diagram of the concept of the proposed multi-antenna system. Using beam-combining technology of the different beams of the patch and monopole antennas, a reconfigurable beam of the proposed multi-antenna system with a feeding network for phase control is obtained.

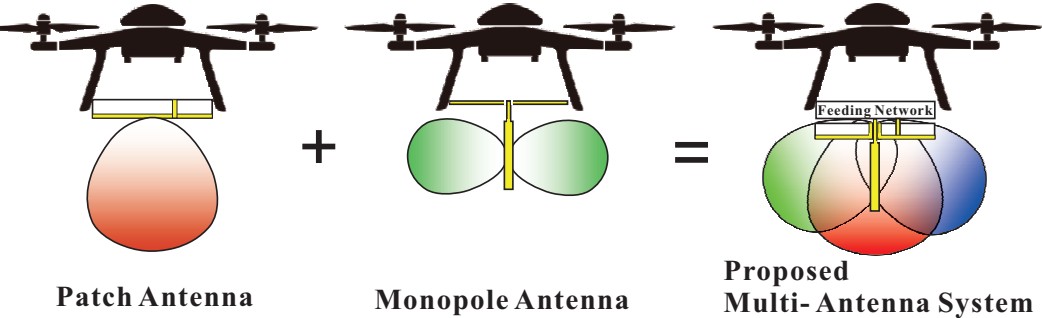

**Figure 1.** The conceptual diagram of the proposed antenna for UAV applications.

*2.1. Radiated Field Analysis by Beam-Combining Technology*

In order to analysis a radiated electric field in far-field region by beam-combining technology, we assume that the coupling between different antennas (quarter-wave monopole and microstrip patch antennas) with a high isolation is neglected. In far-field region ($r > 2D^2/\lambda$ or $r > 3\lambda$), the electric field of the quarter-wave monopole antenna correlates to half-wave dipole antenna [15]. The radiated electric field ($\mathbf{E}_m(r, \theta, \phi)$) of the quarter-wave monopole antenna is as follows:

$$\mathbf{E}_m(r, \theta, \phi) = \hat{\theta}j\eta\frac{I_0 e^{-jkr}}{2\pi r}\left[\frac{\cos(\frac{\pi}{2}\cos\theta)}{\sin\theta}\right] = \hat{\theta}E_{\theta, m}^0 e^{-jkr}. \tag{1}$$

The radiated electric field ($\mathbf{E}_p(r, \theta, \phi)$) of a circular microstrip patch antenna is as follows:

$$\begin{aligned}\mathbf{E}_p(r, \theta, \phi) &= -\hat{\theta}j\frac{k_0 a_e V_0 e^{-jk_0 r}}{2r}\{\cos\phi J'_{02}\} + \hat{\phi}j\frac{k_0 a_e V_0 e^{-jk_0 r}}{2r}\{\cos\theta\sin\phi J'_{02}\} \\ &= \hat{\theta}E_{\theta, p}^0 e^{-jkr} + \hat{\phi}E_{\phi, p}^0 e^{-jkr}\end{aligned} \tag{2}$$

where $a_e$ ($a_e = a\{1 + \frac{2h}{\pi a \varepsilon_r}[\ln(\frac{\pi a}{2h}) + 1.7726]\}^{1/2}$, $a$ = actual radius, $h$ = substrate height, $\varepsilon_r$ = the dielectric constant of the substrate) is an effective radius of the circular patch, $V_o$ is a magnitude of voltage induced in the antenna, and $J$ is the current density, which means the amount of current per unit volume of the antenna ($J'_{02} = J_0(k_0 a_e \sin\theta) - J_2(k_0 a_e \sin\theta)$, and $J_{02} = J_0(k_0 a_e \sin\theta) + J_2(k_0 a_e \sin\theta)$). The beam-combined total electric field ($\mathbf{E}_T(r, \theta, \phi)$) of the proposed multi-antenna system with a relative phase difference ($\phi_0$) between antennas can be obtained by

$$
\begin{aligned}
\mathbf{E}_T(r, \theta, \phi) &= \mathbf{E}_m(r, \theta, \phi)\, e^{j\phi_0} + \mathbf{E}_p(r, \theta, \phi) \\
&= \hat{\theta} E_{\theta, m}^0 e^{-jkr} e^{j\phi_0} + \hat{\theta} E_{\theta, p}^0 e^{-jkr} + \hat{\phi} E_{\phi, p}^0 e^{-jkr}.
\end{aligned}
\tag{3}
$$

By adjusting $\phi_0$ (0° or 180°), the beams in the multi-antenna system can be controlled. Therefore, we can achieve a beam-reconfigurable antenna.

## 2.2. Antenna System Configurations

Figure 2a shows the configuration of the proposed antenna with the feeding network in this paper. The proposed antenna system consists of a meandered monopole antenna, a circular patch antenna, and a coplanar waveguide with ground (CPW-G) structured feeding network. The monopole antenna in Figure 2b was designed in the form of a printed meandered monopole antenna with a 15% reductions in height compared to a $1/4\ \lambda_0$ monopole antenna at 5.09 GHz to achieve low height for UAV communications applications. Furthemore, it connects to the feeding network using P1. Figure 2c has a circular patch antenna with a clearance for monopole feeding at the center, and two feeders P2 and P3 separated by 90 degrees from the center. The two feeders are designed to implement −45° linear polarization and +45° linear polarization, respectively. The two antennas use the ground plane of the feeding network connected to the lower part as a ground plane, and each size is as follows. $H_m$ = 11 mm, $W_s$ = 7.2 mm, $W_{mt}$ = 6 mm, $W_{mb}$ = 0.7 mm, $W_{ms}$ = 3.6 mm, $L_{ms}$ = 0.5 mm, $r_p$ = 8.95 mm, $D_p$ = 2.4 mm, $r_{ps}$ = 15 mm, and $r_f$ = 29.47 mm. The electrical sizes of the proposed antenna and the feeding network at 5.09 GHz are approximately $0.51\lambda_0 \times 0.51\lambda_0 \times 0.2\lambda_0$ and $\lambda_0 \times \lambda_0 \times 0.013\ \lambda_0$, respectively. The total weight is approximately 19.2 g. As a result Figure 2d shows the fabricated multi-antenna. The configuration of the feeding network is shown in Figure 3a–c. It has a CPW-G structure to reduce the path loss at high frequencies. It consists of two input ports, a 90° hybrid coupler, the single-pole double-throw (SPDT) switch, and microstrip 90° phase delay line. The feeding line of the CPW-G structure has 50 Ω impedance when $s$ = 1.47 mm and $w$ = 0.5 mm at 5.09 GHz. As a result, Figure 3d shows the fabricated feeding network. In Figure 3b, c, a microstrip 90° phase delay line has approximately 41° because a phase delay of about 49° occurred in SPDT switch. Taking this into account, the feeding network was constructed, and four modes were implemented to radiate the beam in four directions by adjusting power and phase difference to meandered monopole and circular patch antennas. For circular patch polarization control, the SPDT switch states (P2 and P3) are controlled through $VC1$ and $VC2$ in Figure 3d. When high (3 V) and low (0 V) DC voltages are applied to SPDT $VC1$ and $VC2$, respectively. RF current entering the SPDT switch flows through P2. If low (0 V) and high (3 V) DC voltages are applied to $VC1$ and $VC2$ in the same way, RF current flows through P3.

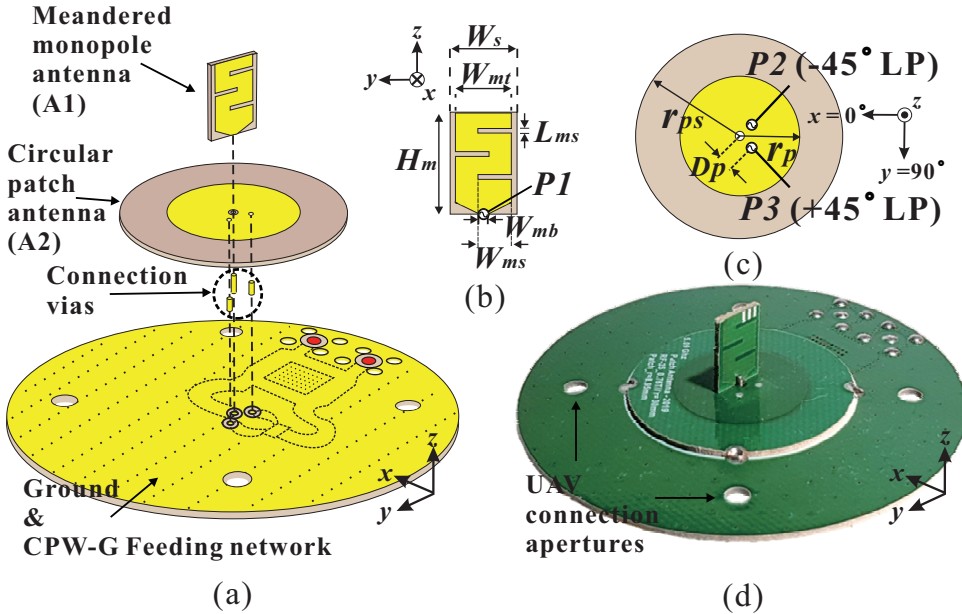

**Figure 2.** Configuration of proposed multi-antenna and its fabricated prototype: (**a**) perspective view of the proposed multi-antenna with the feeding network, (**b**) meandered monopole antenna, (**c**) top view of the circular patch antenna, and (**d**) the proposed multi-antenna prototype.

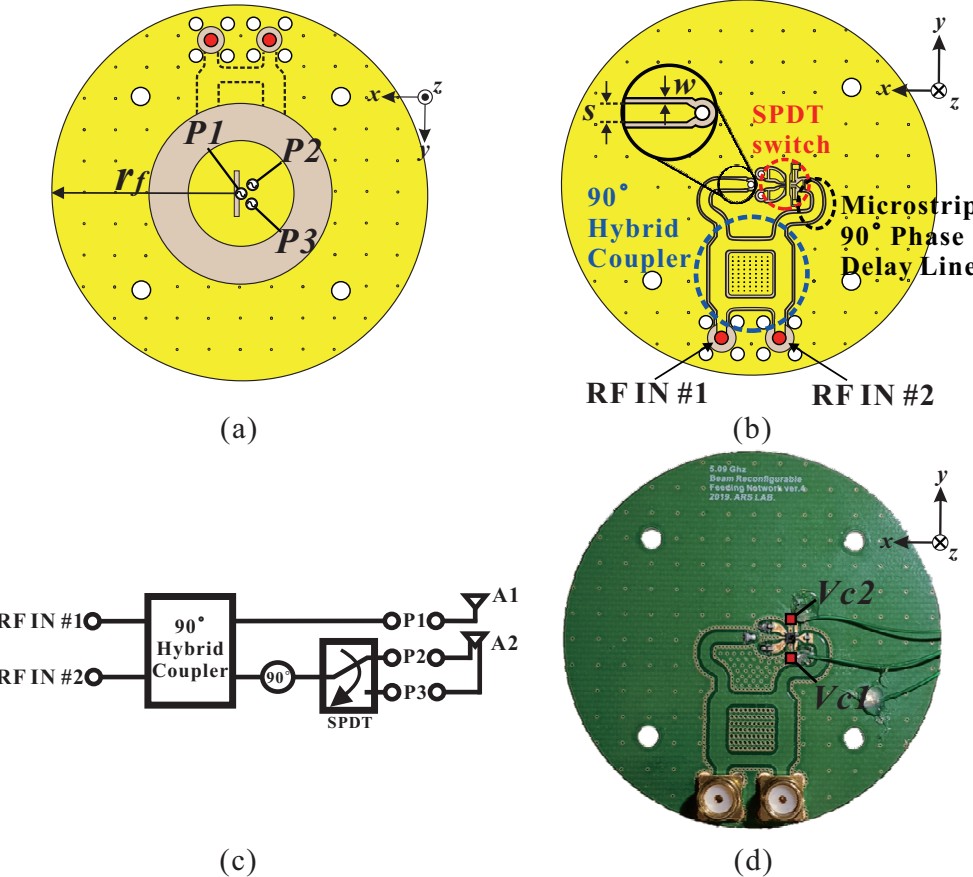

**Figure 3.** Configuration of proposed the feeding network and its fabricated prototype: (**a**) top view of the multi-antenna and feeding network, (**b**) bottom view of the multi-antenna and feeding network, (**c**) the proposed feeding network for four beam pattern using power dividing, and (**d**) phase differences and bottom view of the fabricated multi-antenna and feeding network.

Using the current adjustment of the SPDT state P2 ($-45°$ linear polarization (LP)), the half-power and $180°$ phase difference from RF IN #1 are applied to A1 (meandered monopole antenna) and A2 (circular patch antenna) for the directive broadside beam of Mode 1. This beam radiates in the direction of $\phi = 45°$ and $\theta = 30°$. In the case of the input of RF IN# 1 and SPDT state P3 ($+45°$ LP), A1 and A2 are applied with a half power and a phase difference of $180°$, respectively, and the directive broadside beam of Mode 2 is formed. This beam radiates in the direction of $\phi = 310°$ and $\theta = 30°$. When the SPDT is connected to P2 in RF IN #2, the half-power with same phase is fed to A1 and A2. In Mode 3, the directive broadside beam in the direction of $\phi = 225°$ and $\theta = 30°$ is formed.

Mode 1 and Mode 3 are implemented in A2 through P2, which is a $-45°$ LP, and the phase difference between these two modes shows that the beams radiate in opposite directions. This is the same in Mode 2 and Mode 4 implemented through P3. Thus, when the half power with the equal phase delay is applied to A1 and A2 through the input of RF # 2 and P3 connection, the directive broadside beam of Mode 4 is formed, which is in the direction of $\phi = 135°$ and $\theta = 35°$. Regardless of the input modes, P1 is always on and the power distribution and phase difference at different modes are shown in Table 1.

**Table 1.** Amplitude and phase distributions with SPDT states.

| Distributions | | Input (P1 Is Always on) | | | |
|---|---|---|---|---|---|
| (Ampli., Phase) | | RF IN #1 | | RF IN #2 | |
| Output | A1 | $1/\sqrt{2}, 0°$ | | $1/\sqrt{2}, 90°$ | |
| | A2 | $1/\sqrt{2}, 180°$ | | $1/\sqrt{2}, 90°$ | |
| SPDT States | | P2 | P3 | P2 | P3 |
| Mode | | Mode1 | Mode2 | Mode3 | Mode4 |

## 3. Results and Discussions

The proposed antenna with the feeding network was optimized using a full wave commercial electromagnetic tool (CST Microwave Studio 2019). It was fabricated by TACONIC RF-35 substrate ($\varepsilon_r = 3.5$, $\delta = 0.0018$) for light weight, compact size, and high radiation efficiency. Its thickness is 0.76 mm and the copper thickness is 18 $\mu$m. The SPDT switch for a mode selection of the feeding network is CG2176X3 by California Eastern Laboratories (CEL).

Figure 4 shows the performance of the proposed multi-antenna system with regard to the input ports. The measured reflection coefficients at Mode 1 is described in Figure 4a. Based on the 10-dB impedance bandwidth, it covers the UAV operating frequencies from 5 GHz to 5.5 GHz. Figure 4b shows the insertion loss for each mode. At 5.09 GHz, the minimum and maximum of the insertion loss within the operating frequency band are approximately 3.4 dB and 4.7 dB, respectively. They include a power divider loss of 3 dB, a SPDT switch insertion loss of 1 dB, and a path loss of microstrip line. Figure 4c shows the measured phase variations with regard to the RF INs. The phase difference between modes 1 and 2 at 5.09 GHz is approximately $180°$, and the phase difference between modes 3 and 4 is approximately $0°$. Figure 4d describes the measured peak gain and total efficiency of each mode with regard to the frequency band. All four modes have the highest performance at 5.09 GHz, and the maximum gain and total efficiency are approximately 5.8 dBi and 76%, respectively.

Figure 5 shows the simulated and measured results of the radiation patterns at each mode in the elevation planes ($+45°$ and $-45°$). In Figure 5a, the boresight direction of Mode 1 is $\theta = 30°$ in the $+45°$ elevation plane. Peak gain and HPBW at that mode are approximately 5.8 dBi and $72.5°$, respectively. In the $+45°$ elevation plane, the radiation pattern at Mode 3 is shown in Figure 5c, and the boresight direction of $\theta$, peak gain and HPBW are approximately $30°$, 5 dBi, and $77.5°$, respectively. Mode 1 and Mode 3 radiate in opposite directions from the azimuth plane. The radiation pattern at Mode 2 is described in Figure 5b. The boresight direction, peak gain and HPBW on the $-45°$ elevation

plane are approximately 30°, 5.7 dBi, and 72.5°, respectively. In Figure 5d, the radiation pattern at Mode 4 is described. The peak gain and HPBW on the −45° elevation plane are approximately 35°, 5.4 dBi, and 77.5°, respectively. Modes 2 and 4 radiate in opposite directions from the azimuth plane. Figures 6 and 7 show measured results of radiation patterns at four modes. Figure 6 shows simulated and measured three-dimensional (3D) radiation patterns for each mode. There was good agreement between the simulated and measured results of the implemented proposed antenna system. Figure 7 shows the measured radiation patterns with regard to the azimuth plane at the input modes. Figures 6a and 7a show the radiation pattern of Mode 1, the peak gain is approximately 5.8 dBi at $\theta$ = 30° and phi = 45°, the HPBW ($\theta$) is approximately 72.5° from −2.5° to 70°, and the HPBW ($\phi$) is 165° from 317.5° to 122.5°. The total efficiency of Mode 1 is approximately 76%. Figures 6b and 7b show the radiation pattern of Mode 2. The peak gain is approximately 5.7 dBi from $\theta$ = 30° and $\phi$ = 310°, the HPBW ($\theta$) is 72.5° from –2.5° to 70°, and the HPBW ($\phi$) is 165° from 242.5° to 47.5°. The total efficiency of Mode 2 is 74%. Similarly, Figures 6c and 7c show the radiation pattern of Mode 3. The peak gain is 5 dBi at $\theta$ = 30° and $\phi$ = 225°, the HPBW ($\theta$) is 77.5° from −2.5° to 75°, and the HPBW ($\phi$) is 190° from 135° to 325°. The total efficiency of Mode 3 is 70.5%. Furthermore, Figures 6d and 7d show the radiation pattern of Mode 4. The peak gain is 5.4 dBi at $\theta$ = 35° and $\phi$ = 135°, the HPBW ($\theta$) is 72.5° from −2.5° to 75°, and the HPBW ($\phi$) is 160° from 47.5° to 207.5°. Total efficiency of Mode 3 is approximately 73.3%. From these results, four modes show a beam-reconfigurable characteristic in which beam is tilted about 90° according to mode change. The minimum gain and total efficiency of four modes, the HPBW ($\theta$), and the HPBW ($\phi$) are 5 dBi, 70.5%, 72.5°, and 160°, respectively.

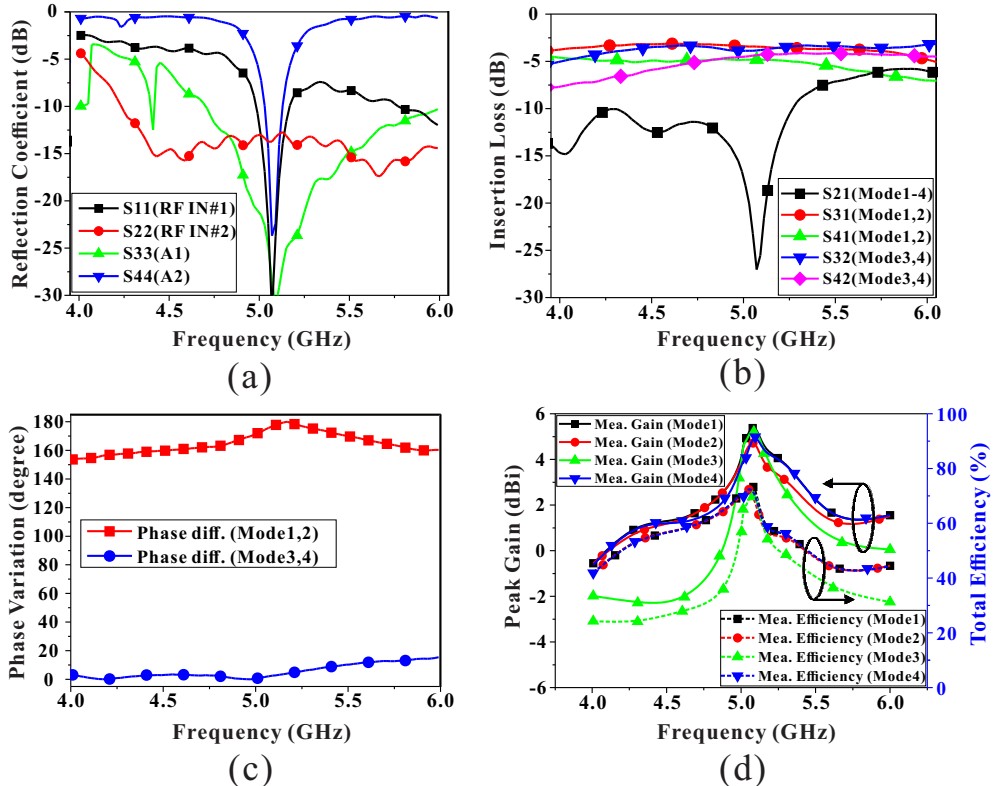

**Figure 4.** Performance of the proposed multi-antenna system with regard to the input modes (1–4): (**a**) measured reflection coefficients, (**b**) measured insertion loss, (**c**) measured phase variations with regard to the RF INs, and (**d**) measured peak gain and total efficiency

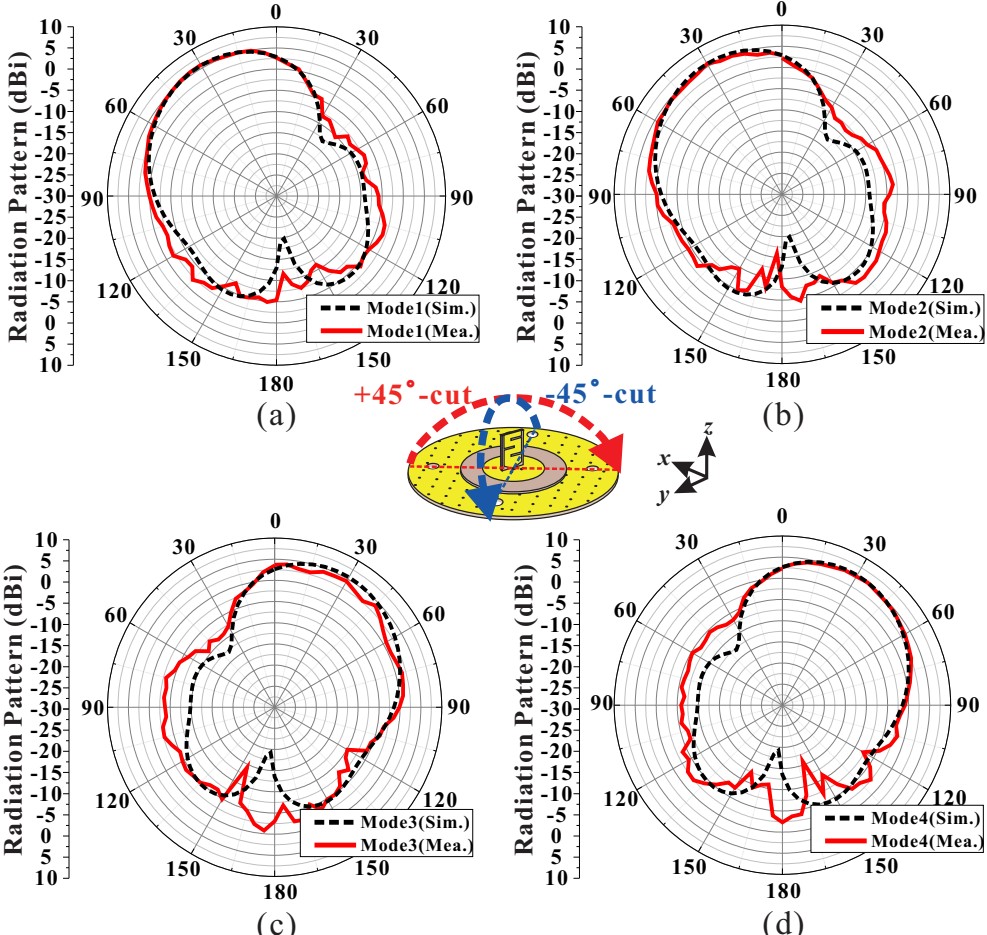

**Figure 5.** Simulated and measured radiation patterns at the input modes: (**a**) Mode 1 (+45° elevation plane), (**b**) Mode 2 (−45° elevation plane), (**c**) Mode 3 (+45° elevation plane), and (**d**) Mode 4 (−45° elevation plane).

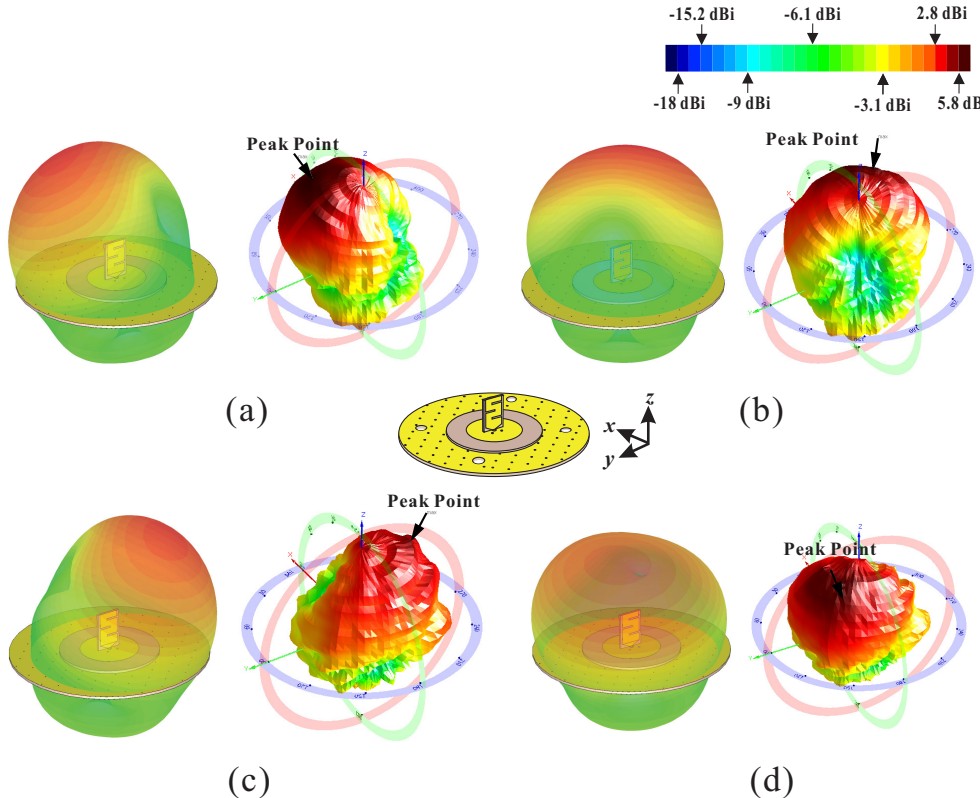

**Figure 6.** Comparison of simulated and measured 3D radiation pattern results for each mode: (**a**) Mode 1, (**b**) Mode 2, (**c**) Mode 3, and (**d**) Mode 4.

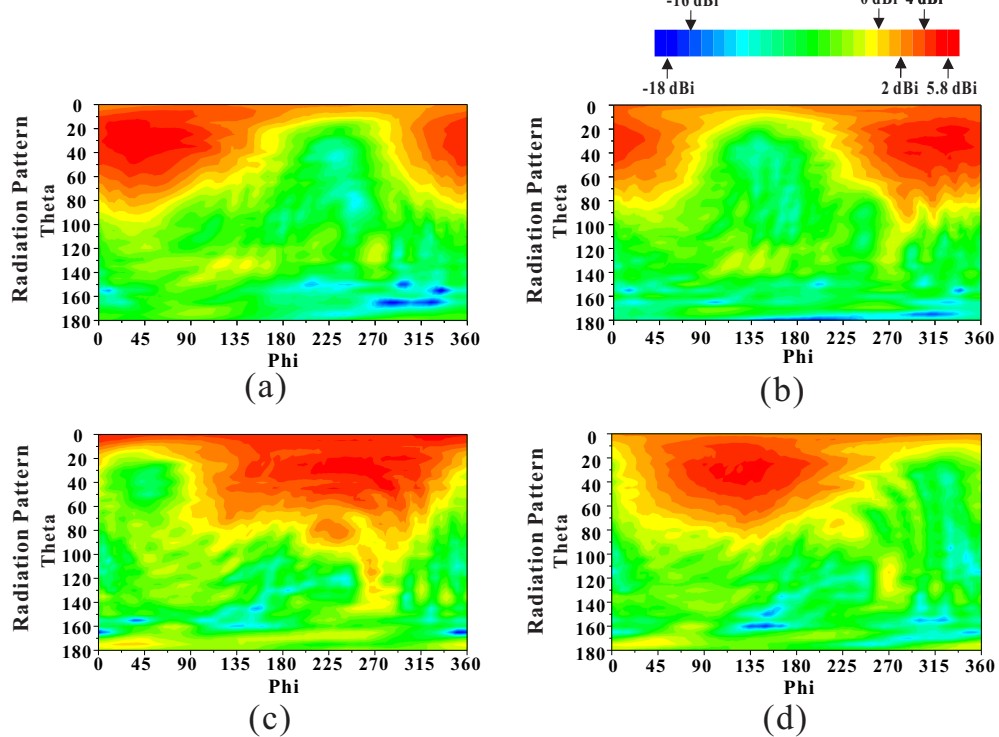

**Figure 7.** Measured radiation patterns with regard to the azimuth plane at the input modes: (**a**) Mode 1, (**b**) Mode 2, (**c**) Mode 3, and (**d**) Mode 4.

Figure 8 shows the synthesis results for each mode of the measured radiation pattern. Figure 8a and Figure 8b show the combined results of the measured radiation patterns at 5.09 GHz in *zx*-plane

and *zy*-plane, respectively. In Figure 8c, the directive broadside beams of each mode cover the whole azimuth plane when θ = 30°. As a result, Figure 8d shows the combined radiation pattern at the azimuth direction (φ) has a wide HPBW elevation angles of approximately 145° from −75° to +70°. A comparison of the previous works with the proposed antenna is shown in Table 2. Most antennas for UAV applications [3–10] use ISM bands rather than UAV frequency band. Ref. [3] has a large size and small gain, and due to the conical beam characteristics, null occurs near the elevation angle of 0°. In the case of [6,7], it operates in the UAV frequency band, but it covers only a wide elevation angle without azimuth angle. In the case of [8], the array of the feeding network and antenna for high gain, the beam coverage is limited, and Ref. [9] combines antennas with different beams to cover a wide azimuth angle, but elevation angle has a narrow coverage feature. Ref. [10] has a high gain, relatively wide 10-dB BW, and wide beam coverage. However, due to the beam tilted characteristics to the physical arrangement, a null zone may exist near the elevation angle 0° from the center of the antenna, resulting in a fade zone. In addition, the box-type structure of the physical arrangement has a large volume, which is disadvantageous for aerodynamics. Refs. [11–14] show a beam switching antenna with a wide coverage, not a UAV antenna. It uses a pin diode [11–13] or switches the beam through the configuration of feeding network [14]. In the case of [11], it has a relatively narrow coverage compared to the proposed antenna, and it is large and heavy due to a metal substrate. Refs. [12,13] have high gain, but also a feature that covers only the azimuth plane, and Ref. [14] forms four beams through the feeding network, but it is large and covers only a wide elevation angle.

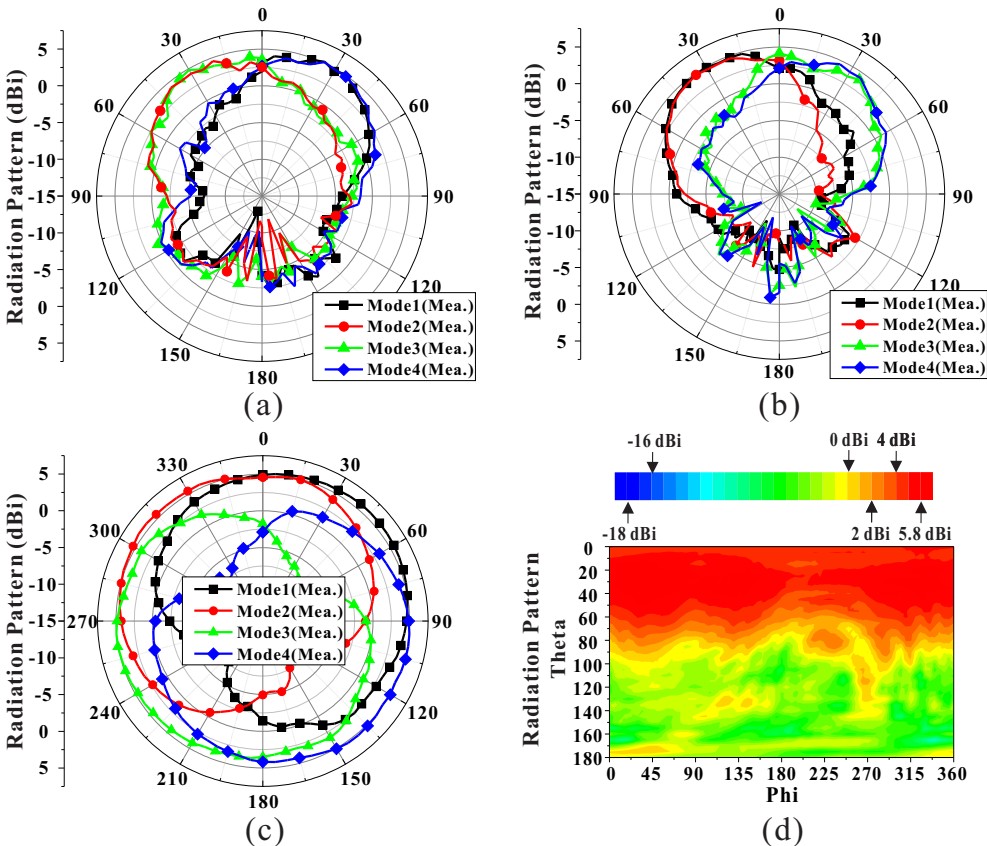

**Figure 8.** Measured radiation patterns and combined results of four input modes at 5.09 GHz: (**a**) *zx*-plane, (**b**) *zy*-plane, (**c**) azimuth plane at θ = 30°, and (**d**) combined radiation beam pattern of four input modes.

**Table 2.** Performance comparison between the previous works and the proposed array about beam-reconfigurable antennas

| Ref. | Freq (GHz) | 10-dB BW (MHz) | Peak Gain (dBi(c)) | HPBW ($\theta$) | HPBW ($\phi$) | Size ($\lambda_0 \times \lambda_0 \times \lambda_0$) |
|---|---|---|---|---|---|---|
| [3] | 2.4 | 100 | 4.3 | 30° + 30° | 360° | 2 × 0.4 × 0.06 |
| [4] | 0.51 | 11 | 4.8 | 110° | 100° | 0.27 × 0.08 × 0.05 |
| [5] | 1.435 | 13 | 1.01 | - | 360° | 0.16 × 0.16 × 0.03 |
| [6] | 5.09 | 850 | 5 | 161° | 160° | 0.95 × 0.9 × 0.34 |
| [7] | 5.09 | 120 | 4.37 | 200° | 160° | 0.81 × 0.68 × 0.36 |
| [8] | 9.84 | 970 | 9.8 | 28° | 360° | 0.89 × 0.22 × 3.05 |
| [9] | 2.4 | 300 | 8 | - | 360° | 0.96 × 0.58 × 0.01 |
| [10] | 5.9 | 1030 | 7.4 | 65° + 65° | 360° | 0.88 × 0.88 × 0.83 |
| [11] | 2.55 | 600 | 7 | 120° | 360° | 1.45 × 1.45 × 0.12 |
| [12] | 2.45 | 550 | 6.5 | - | 290° | 0.57 × 0.45 × 0.28 |
| [13] | 5.5 | 800 | 10 | - | 360° | 2.5 × 2.5 × 0.5 |
| [14] | 2.4 | 83 | 8.3 | 120° | - | 1.98 × 1.16 × 0.48 |
| **Prop.** | **5.09** | **550** | **5.8** | **145°** | **360°** | **1 × 1 × 0.21** |

On the other hand, the proposed antenna was designed for UAV frequency band, and the synthesis of four mode beams can be tilted by approximately 90° electronically with the configuration of the feeding network. The HPBW not only has a wide vertical coverage of 145° and omnidirectional azimuth coverage of 360° but also has a compact size and low height, making it suitable for robust U2X communication implementation in UAV applications.

## 4. Conclusions

We proposed a beam-reconfigurable multi-antenna system with beam combining technology for U2X systems. The proposed antennas with different beams were reconfigured into four-direction beams by controlling the input power and phase difference at the feeding network. The combined beam can be used in various ways by covering a wide coverage. The 10 dB-bandwidth of these beams ranges from 5 GHz to 5.5 GHz, covering all UAV operating frequency bands (5.03–5.15 GHz). The peak gain, the total efficiency, and HPBW ($\theta$, $\phi$) are approximately 5.8 dBi, 76%, 145°, and 360° respectively. Since the proposed multi-antenna system has a compact size, a light weight of 19.2 g, a centralized weight balance, and a low height, it is suitable for a robust U2X system using UAV that is highly applicable to various fields.

**Author Contributions:** Y.-S.C. contributed conceptualization, simulation, analysis, measurements and writing—original draft preparation.; J.-S.P. contributed formal analysis, and fabrications.; W.-S.L. contributed the idea, writing—review and editing, supervision, and overall research. All authors have read and agreed to the published version of the manuscript.

**Funding:** This work was supported in part by the Korean Government (MSIT) through the National Research Foundation of Korea, South Korea, under Grant 2019R1C1C1008102 and in part by the Human Resources Program in Energy Technology of the Korea Institute of Energy Technology Evaluation and Planning funded by the Ministry of Trade, Industry and Energy, South Korea, under Grant 20174030201440.

**Conflicts of Interest:** The authors declare no conflict of interest.

## Abbreviations

The following abbreviations are used in this manuscript:

| | |
|---|---|
| BVLoS | beyond visual line of sight |
| CEL | California Eastern Laboratories |
| CPW-G | coplanar waveguide with ground |
| LP | linear polarization |
| UAV(s) | unmanned aerial vehicles |
| U2X | UAV to everything |
| SPDT | single-pole double-throw |

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
