# Peer review of "Beam-Reconfigurable Multi-Antenna System with Beam-Combining Technology for UAV-to-Everything Communications"

_electronics, doi:10.3390/electronics9060980_

Round 1
Reviewer 1 Report
The authors present a system combining two antennas, a circular patch antenna and a monopole, with a feeding system to provide wide beam coverage. The concept and results are interesting. However, there are a number of concerns that need to be carefully addressed by the authors.
First of all, the authors need to better highlight the novelty of their work in the introduction. They also need to improve the review and analysis of other works in the literature on the topic. With regard to other works, what is the novel contribution of this work?
Regarding Fig. 4d, please explain the graph, since there are two antennas and four modes of operation but only one peak gain and efficiency figure.
It is possible that the 3D radiation patterns in Figs. 6 and 7d may be better represented in U-V coordinates.
According to Table 2, the results in [10] are better than in the present manuscript, and those in [11] are only slightly behind. Thus, it is not clear what is the novel contribution of this work. Despite the good results, it seems that this work does not have added academic value.
Author Response
Thanks for giving us an opportunity to improve our manuscript. In the revised manuscript we have included additional explanations, discussions and corrections, closely addressing all the comments and suggestions pointed out by the reviewers. All the changes made have been highlighted in the revised manuscript.
We would like to express our sincere appreciation of the reviewers’ comments regarding this paper. Our responses are attached.
Sincerely yours,
Wang-Sang Lee

Reviewer 2 Report
The presented technique sounds very well. The manuscript is acceptable for publication.
Author Response

(The authors gave the same response as above.)

Reviewer 3 Report
Interesting paper, but it needs to be revised so that it can be understood properly.
1
The abstract says: “This paper proposes a beam-reconfigurable antenna for unmanned aerial vehicles (UAVs) with wide beam coverage by applying beam-combining technology to multiple antennas with different beam patterns.”
According to the description of the paper it seems to be only 2 antennas, a monopole, and a patch. Do the authors intend to build some kind of array? They should clarify.
2
Some figures have text in a disproportionate size compared to the flowing text. They must correct.
3
In lines 50 and 51 in the description of figure 2c, refer to two feeders P2 and P3 that are only defined in figure 3.
4
In line 52 the phrase "…to implement -45 linear polarization and +45 linear polarization…", is not clear. Polarization is at 45 with respect to what? As we know in a 3D space, an angle is not enough to define a direction, much less without reference.
5
What do the quantities, (s = 1.47 mm, w = 0.5 mm, 50 W), described in line 60 mean?
6
From here, deductions and conclusions are impossible to obtain based exclusively on the microstrip line, with an electrical length of 90 connected to the SPDT. What is necessary to know is the difference between the electrical length between this line and the line on the left, which seems to be higher. What is wrong?
7
Based on the previous point, it seems impossible to have a 180º difference between the signals at A1 and A2. This point is central to understanding the entire performance of this system. Most of the deductions and conclusions are based on the fact that the microstrip line that connects A1 has zero electrical length, so it will be wise to carefully analyze this point and adjust the rest of the paper.
Other suggestions, in case the authors clarify point 7.
8
A 3D Radiation Diagram Pattern, for at least one situation it is highly desirable to understand the obtained diagram. Why is the "45-cut" plane (a vertical plane?) Called "45 elevation plane?"
9
in figure 4b) the measurement S21 (although it is measured in the same way as the others) is normally known as isolation and should have the "attenuation" as high as possible. In this sense, it should be removed, allowing the remaining measures to be observed on a more favorable scale, where a few tenths of dB can make a project unfeasible.
Author Response

(The authors gave the same response as above.)

Reviewer 4 Report
The authors proposed a beam-reconfigurable multi-antenna system with beam combining technology for UAVs. By controlling the input power and phase difference at the feeding network, they developed four beams and combine them together to reach a competitive antenna design. Here, I have some suggestions for the authors.
1) More details about how to control the phase, i.e., the switch.
2)More theoretical analysis on the radiation patterns are expected, i.e., why the patterns look like Fig.5,6,7.
3) Compared with [10] in table 2-2, is your design better than the referred one?
Author Response

(The authors gave the same response as above.)

Round 2
Reviewer 1 Report
The authors have improved the paper by addressing the concerns of the reviewer. However, after doing so, I have some new concerns.
The introduction has been improved by commenting pros and cons of other works in the literature on this topic. However, the two lines dedicated to their own work is not enough and authors should expand on their work compared to others broadly. They can go on details at the end of the manuscript.
Since now you have introduced equations in Section 2.1, the parameters and functions should be described. For instance, the J functions, a_e, phi_0, etc.
Reviewer 3, concern 4: on the +-45 linear polarizations, I think the paper would improve if a graphical explanation is added, since as the reviewer 3 points out, we need to have the reference for the angle. I think it could be easily added to existing Fig. 2 to one of the subfigures.
Page 1, line 4: combing -> combining
Page 9, line 151: tiltied -> tilted
Author Response
Thanks for giving us an opportunity to improve our manuscript. In the revised manuscript we have included additional explanations, discussions and corrections, closely addressing all the comments and suggestions pointed out by the reviewers. All the changes made have been highlighted in the revised manuscript.
Sincerely yours,
Wang-Sang Lee

Reviewer 3 Report
My main issue has not been clarified.
Although it seems to me that the gap between A1 and A2 that the authors say is 180, in modes 1 and 2, and 0 in modes 3 and 4, does not influence the result obtained, I think that the paper could be clarified.
Transcribing:
C.7. Based on the previous point, it seems impossible to have a 180º difference between the signals at A1 and A2. This point is central to understanding the entire performance of this system. Most of the deductions and conclusions are based on the fact that the microstrip line that connects A1 has zero electrical length, so it will be wise to carefully analyze this point and adjust the rest of the paper.
Other suggestions, in case the authors clarify point 7.
RESPONSE: It supposes that all electrical lengths are the same except for the 90° microstrip line in Figure 3c. Thus, when entering the signal in RF #1 in Figure 3c, A1 has a relatively 0° phase, A2 adds a 90° phase through the 90° hybrid coupler, and the total 180° phase changes through the 90° delay line. Therefore, the relative phase difference between A1 and A2 is 180°. Similarly, when signals are applied to RF #2, A1 has a 90° phase through a 90° phase change in a 90° hybrid coupler, and A2 has a 90° phase through a 90° delay line, resulting in a 0° relative phase difference between A1 and A2."
The "RESPONSE" would be correct if A1 was directly at the hybrid port, but between the hybrid port and A1 there is an undisclosed electrical length line that makes the answer given at least scientifically doubtful.
Author Response

(The authors gave the same response as above.)
